# Capturing Failures of Large Language Models via Human Cognitive Biases

**Erik Jones**
UC Berkeley
erjones@berkeley.edu

**Jacob Steinhardt**
UC Berkeley
jsteinhardt@berkeley.edu

## Abstract

Large language models generate complex, open-ended outputs: instead of outputting a class label they write summaries, generate dialogue, or produce working code. In order to asses the reliability of these open-ended generation systems, we aim to identify qualitative categories of erroneous behavior, beyond identifying individual errors. To hypothesize and test for such qualitative errors, we draw inspiration from human cognitive biases—systematic patterns of deviation from rational judgement. Specifically, we use cognitive biases as motivation to (i) generate hypotheses for problems that models may have, and (ii) develop experiments that elicit these problems. Using code generation as a case study, we find that OpenAI's Codex errs predictably based on how the input prompt is framed, adjusts outputs towards anchors, and is biased towards outputs that mimic frequent training examples. We then use our framework to elicit *high-impact errors* such as incorrectly deleting files. Our results indicate that experimental methodology from cognitive science can help characterize how machine learning systems behave.[1]

## 1 Introduction

Recent large language models have achieved new, exciting capabilities. In contrast to traditional classifiers, these models can generate open-ended text, enabling use cases like summarization [Stiennon et al., 2020], dialog [Thoppilan et al., 2022], and code generation [Chen et al., 2021].

The open-ended power of these systems, however, poses new reliability challenges. We must understand not only when systems err, but also the *kinds* of errors they make, as some errors are much more costly than others. For example, erroneous code that does not compile is less dangerous than code that deletes all files in the home directory. Studying how frequently an error occurs is difficult, as the same error (e.g. delete all files) can appear in a wide range of syntactically diverse outputs. In order to better reason about how complex systems err, we need methods to test whether systems make the same qualitative error across different prompts, even when the generated outputs differ.

To study these reliability challenges, we primarily focus on code generation models. Such models complete programs from comments, descriptions of code functionality, or initial lines of code. Code generation is particularly amenable to study since it is *objective*: generated solutions are unambiguously correct or incorrect. Yet it is also *open-ended*: the set of programs a model could output is arbitrarily large, so the rate at which a specific program is outputted is not very descriptive.

Many of the reliability challenges posed by code generation models, and open-ended systems broadly, also arise when studying qualitative failures in human decision making. These failures, called *cognitive biases*, are systematic ways in which humans deviate from rational judgment [Tversky and Kahneman, 1974]. For example, Tversky and Kahneman find that humans inadequately adjust

---

[1]Code for this paper is available at `https://github.com/ejones313/codex-cog-biases`.

36th Conference on Neural Information Processing Systems (NeurIPS 2022).

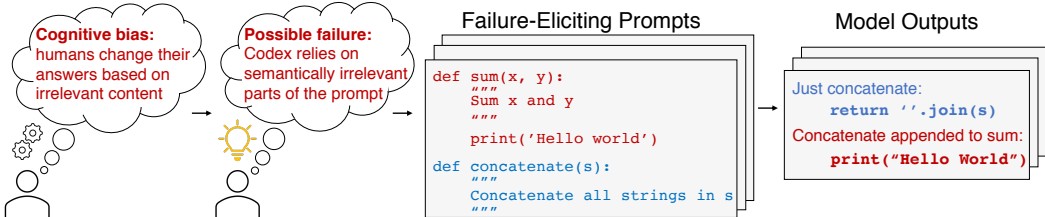

Figure 1: Illustration of our experimental framework. We use a cognitive bias (framing effect) to inspire a potential code generation failure mode (relying on irrelevant information). We then transform inputs in a way that we suspect will elicit the failure mode (prepending sum). We evaluate whether the modifications lower accuracy, and if the output is an instance of the targeted failure mode.

estimates away from initial values, and disproportionately recall distinctive examples. To uncover cognitive biases, Tversky and Kahneman ask questions that are crafted to systematically reveal some qualitative irrationality. They uncover insights into human behavior from the diverse responses, without complete mechanistic insight into the minds that they aim to analyze.

In this work, we extend Tversky and Kahneman's experimental methodology and results to elicit failure modes of large code and language models, without relying on complete mechanistic insight into their behavior (Figure 1). Given a potential failure mode (e.g. relying on irrelevant information in the input), we construct a transformation over inputs that largely preserves semantics, but that we suspect will elicit the failure (e.g. prepending an irrelevant function). We first test if the model is sensitive to the transformation, by measuring if it decreases accuracy. Then, we check that the model outputs have elements that are indicative of the targeted failure (e.g. copies the irrelevant function).

We draw on four different cognitive biases to hypothesize potential failures of OpenAI's Codex [Chen et al., 2021] and Salesforce's CodeGen [Nijkamp et al., 2022], then apply our framework to each. Our results indicate that these models often rely on irrelevant information when generating solutions, adjust solutions towards related-but-incorrect solutions, are biased based on training-set frequencies, and reverts to computationally simpler problems when faced with a complex calculation. We also apply our framework to OpenAI's GPT-3 [Brown et al., 2020], and show that it updates its predictions towards anchors, and predictably adjusts its responses based on the question framing.

Finally, we show that our framework can uncover *high-impact errors*: errors that are harmful and difficult to undo. Specifically, we use our framework to systematically generate prompts where Codex erroneously deletes files. Our results indicate that experimental methodology from cognitive science can help uncover failure modes of complex machine learning systems.

## 2   Related Work

**Large language models.** Recent work has developed large, capable, autoregressive language models, which predict future tokens from past tokens [Radford et al., 2019, Wang and Komatsuzaki, 2021, Brown et al., 2020, Chen et al., 2021, Rae et al., 2021]. These models can be used for open-ended generation tasks such as summarization [Stiennon et al., 2020, Ziegler et al., 2019, Rothe et al., 2020], dialogue [Ram et al., 2018, Thoppilan et al., 2022], and long form question answering [Fan et al., 2019], among others. Model-generated code has been used to solve both programming and statistics questions [Chen et al., 2021, Tang et al., 2021].

There is some existing work studying failures of large language models. Benchmarks that measure model performance on multiple choice questions [Wang et al., 2019b,a, Hendrycks et al., 2021b], mathematics [Hendrycks et al., 2021c, Cobbe et al., 2021], long-form question answering [Lin et al., 2021, Gabriel et al., 2021, Shuster et al., 2021, Krishna et al., 2021], and coding problems [Hendrycks et al., 2021a, Chen et al., 2021] reveal inputs that the model errs on, but not the kind of error it makes. Another line of work shows that test-based language models can internalize bias and stereotypes [Sheng et al., 2019, Nadeem et al., 2020, Groenwold et al., 2020, Blodgett et al., 2021, Gehman et al., 2020], and proposes applying fairness measurements from cognitive social sciences to machine learning systems [Jacobs and Wallach, 2021]. Some work adversarially prompts models to leak training data [Carlini et al., 2020], or output specific content [Wallace et al., 2019, Carlini et al.,

```
Prompt          ⎡ def largest_divisor(n: int) -> int:          def sum(x, y):                ⎤  Irrelevant
                ⎢     """ For a given number n, find the largest    """                        ⎥  Preceding
                ⎢     number smaller than n that divides n evenly   Returns the sum of x and y  ⎥  Function
                ⎢     >>> largest_divisor(15)                       """                        ⎥
                ⎣     5                                             print('Hello world')       ⎦
                      """

Canonical       ⎡ for i in reversed(range(n)):                  def concatenate(s):
Solution        ⎢     if n % i == 0:                                """
                ⎣         return i                                  Returns the concatenation of all
                                                                    strings in s
Test            ⎡ def check(candidate):                             """
Cases           ⎢     assert candidate(3) == 1                   ─────────────────────────────────
                ⎣     assert candidate(49)== 7                      print('Hello world')
```

Figure 2: **Left.** Example of a HumanEval problem from Chen et al. [2021]. The problem contains a prompt (blue), a canonical solution to the prompt (green), and a few test-cases (black). The prompt contains two components: a function signature (first line), and a docstring (remaining lines). **Right.** Illustration of our framing experiment. The transformed prompt (everything above the black line) contains an irrelevant preceding function (IPF) prepended to a prompt from HumanEval (blue). The IPF contains a randomly chosen prompt from HumanEval (purple) and a framing line (red). The output Codex generates (below the black line) matches the framing line. When we omit the random HumanEval prompt and the framing line (leaving only blue), Codex produces the correct output.

2020]. And a final line of work identifies additional potential failures of current and future machine learning systems [Bender et al., 2021, Bommasani et al., 2021, Weidinger et al., 2021].

**Cognitive biases.** Tversky and Kahneman [1974] define *human cognitive biases*: systematic patterns of deviation from rational judgment. They observe that humans employ heuristics when computing probabilities or assessing values, and that these heuristics lead to predictable errors. Follow-up work has added to, refined, and validated the set of known cognitive biases [Tversky and Kahneman, 1973, 1981, Strack et al., 1988, Kahneman and Frederick, 2002, Windhager et al., 2010, Meyer, 2014].

Some known failure modes of large language models resemble cognitive biases. Zhao et al. [2021] and Liu et al. [2021] show that the specific random samples used for few-shot learning can change GPT-3's prediction on binary and multiple choice tasks. Similarly, Wallace et al. [2019] show that innocuous prompts can routinely generate toxic model output. Our framework builds on this work by (i) identifying the link to cognitive biases, (ii) focusing on open-ended generation, and (iii) leveraging Tversky and Kahneman's experimental methodology to elicit qualitative failure modes.

## 3 Code Generation Experiments

### 3.1 Models

We study two code models: OpenAI's Codex [Chen et al., 2021], and Salesforce's CodeGen. [Nijkamp et al., 2022]. Both models are autoregressive—given a sequence of previous tokens, they predict the next token. Practitioners query these code models with partial programs, docstrings, or function signatures, and obtain completions as output.

**Codex.** We study OpenAI's Codex, a large language model trained to generate code from docstrings [Chen et al., 2021]. We use the OpenAI API to query the "davinci-001" version of Codex, and use greedy decoding to generate solutions. Details of this model architecture are not public, but it is likely similar to the largest model from Chen et al. [2021]: a 12B parameter version of GPT-3 [Brown et al., 2020] that is fine-tuned on GitHub instead of the CommonCrawl.

**CodeGen.** We additionally study the 6.2 billion parameter "mono" version of CodeGen, which is trained on text data and fine-tuned on GitHub. Unlike Codex, the weights of CodeGen are publicly available,[2] so we run inference locally. We use greedy decoding to generate solutions.

### 3.2 Benchmarks

In order to identify whether code models some failure mode, we need to generate prompts that elicit that failure. To do so, we systematically apply transformations to standard prompts. We use two benchmarks as sources of prompts to transform: HumanEval, and MathEquations.

---

[2] https://github.com/salesforce/CodeGen

| Framing Line | Model | Functional accuracy | | Outputs framing line | |
|---|---|---|---|---|---|
| | | ORIGINAL | FRAMED | ORIGINAL | FRAMED |
| `raise NotImplemented` | CODEX | 32.9 | 2.4 | 1.4 | 91.7 |
| | CODEGEN | 25.6 | 1.5 | 0.0 | 79.3 |
| `pass` | CODEX | 32.9 | 3.0 | 9.7 | 92.7 |
| | CODEGEN | 25.6 | 2.1 | 0.0 | 78.7 |
| `assert False` | CODEX | 32.9 | 3.3 | 0.0 | 92.7 |
| | CODEGEN | 25.6 | 4.2 | 0.1 | 72.6 |
| `return False` | CODEX | 32.9 | 4.9 | 11.5 | 65.6 |
| | CODEGEN | 25.6 | 3.6 | 0.0 | 64.6 |
| `print("Hello world!")` | CODEX | 32.9 | 10.6 | 0.0 | 62.2 |
| | CODEGEN | 25.6 | 11.0 | 0.0 | 58.2 |

Table 1: Results of the framing experiments. We compare functional accuracy and the rate at which framing line is outputted over HumanEval with (framed) and without (original) irrelevant preceding functions. We find that the irrelevant preceding functions lower functional accuracy across all framing lines for Codex and CodeGen. Moreover, we find that the outputted function often appears verbatim in the generated output, suggesting that both models rely on irrelevant information in the prompt.

**HumanEval.** We use the HumanEval benchmark as a diverse source of "normal" prompts [Chen et al., 2021]. HumanEval contains 164 programming problems, each of which includes a function signature and a docstring. The docstring contains an English description of the desired functionality and a few example input-output pairs. HumanEval also contains a canonical solution for each program, which we use in Section 3.3.2. We give an example problem from HumanEval in Figure 2.

**MathEquations.** We also curate a set of prompts of basic arithmetic functions. For example, we prompt Codex to "Write a function that sums the squares of its inputs", or "Write a function that sums its inputs called product_plus_five". Further details are given in Sections 3.3.3 and 3.3.4.

### 3.3 Empirical results

In this section, we show how cognitive biases can (i) inspire hypotheses for potential failure modes, and (ii) help us design experiments to test these hypotheses. Our approach has three steps. First, we construct a transformation over prompts that largely preserves semantics, but that we suspect will elicit a specific cognitive-bias-inspired failure mode. Next, we measure if code models are sensitive to the transformation, by measuring the decrease in accuracy. And finally, we check that the generated output has elements that are indicative of the targeted failure mode. Our approach mirrors the high-level methodology from Tversky and Kahneman [1974]; we empirically elicit specific failure modes using targeted prompts, without complete mechanistic insight into the system that we study.

We draw inspiration from four cognitive biases: the framing effect (Tversky and Kahneman [1981]; Section 3.3.1), anchoring (Tversky and Kahneman [1974]; Section 3.3.2), the availability heuristic (Tversky and Kahneman [1973]; Section 3.3.3), and attribute substitution (Kahneman and Frederick [2002]; Section 3.3.4).

#### 3.3.1 Inspiration: Framing effect

We first draw inspiration from *the framing effect*: predictable shifts in human responses when the same problem is framed in different ways [Tversky and Kahneman, 1981]. In their study identifying the effect, Tversky and Kahneman [1981] find that subjects favor certainly saving 200 people over saving 600 with probability 1/3, yet prefer losing 600 with probability 2/3 over certainly losing 400 (even though these are equivalent). At its core, the framing effect shows how humans can rely on semantically irrelevant information when they make decisions.

Using the framing effect as inspiration, we hypothesize that code generation models may generate solutions exclusively from irrelevant information in the prompt. To elicit this failure, we transform HumanEval prompts by prepending *irrelevant preceding functions*. Specifically, to generate irrelevant

```python
def common(l1, l2):
  ret = set()
  for el in l1:
    for var in [l1, l2]:
    print(var)
```

| Anchor Function |
|---|
| HumanEval Prompt (def common...) |
| First solution lines |

| Full Prompt |
|---|
```python
for var in [l1, l2]:
    if el1 in var:
        ret.add(e1)
return sorted(ret)
```

Figure 3: Illustration of our anchoring experiment using a real example (expanded in Figure 7). We construct the anchor function (left) by taking the function signature from the HumanEval prompt (blue), appending $n$ lines of the canonical solution (green), then adding anchoring lines (red). We construct the full prompt (center) by combining the anchor function, the original HumanEval prompt, and the first $n$ lines of the canonical solution. The solution Codex generates (right) combines elements of a canonical solution (checks condition and adds to ret.), with the anchor function (for var loop).

preceding functions, we combine a random prompt from HumanEval with a *framing line*. We test five framing lines: `raise NotImplementedError`, `pass`, `assert False`, `return False`, and `print("Hello world!")`. We first check that prepending these irrelevant preceding functions decreases functional accuracy.[3] Next, to test if models relied on irrelevant information in the prompt, we measure how much more frequently the framing line appears verbatim in the generated output.

We report the results of our framing experiments in Table 1. We find that adding irrelevant preceding functions consistently lowers functional accuracy, by between 22.3 and 30.5 points for Codex, across the different framing lines we tested. Moreover, both models frequently generate the framing line: 81% of the time for Codex and 70.7% of time for CodeGen, compared to only 4.5% and 0.0% over untransformed prompts respectively. These results suggest that code generation models can erroneously rely on irrelevant information in the prompt in predictable ways, even in the extreme case when doing so contradicts the type specification in the function signature (`return False`).

### 3.3.2 Inspiration: Anchoring

We next draw inspiration from *anchoring*: humans' tendency to insufficiently adjust their estimates away from initial values. For example, Tversky and Kahneman [1974] find that subjects' median estimate for the fraction of African countries in the UN shifts from 25% to 45%, based on whether they were first asked if the fraction was greater or less than 10% and 65%, respectively. Anchoring captures how humans adjust to partial information, versus irrelevant information (framing effect).

Using anchoring as inspiration, we hypothesize that code generation models may adjust their output towards related solutions, when these solutions are included in the prompt. To elicit this failure, we prepend *anchor functions* to prompts: functions that are similar to a valid solution for a HumanEval prompt, but contain some error. We first check that prepending these anchor functions decreases functional accuracy, as in Section 3.3.1. Next, to test if models adjust their output towards related solutions, we check that the generated solution contains elements of the anchor function.

We aim to construct anchor functions that are similar to functions in HumanEval prompts and that compile, but are incorrect. To do so, we take a prefix of the canonical solution, then add additional *anchor lines* that produce an incorrect output. See Figure 3 for an example. We describe two types of anchor lines, and how we test their influence on the generated solutions, in the following paragraphs.

**Print-var anchor lines.** We first study *print-var* anchor lines, which iterate over all variables in the function signature and print their values. For a function with inputs `var1` and `var2`, the associated print-var anchor lines are:

```python
for var in [var1, var2]:
    print(var)
```

To study the influence of the print-var anchor lines on the solution, we measure how often (i) just the first line (for loop), and (ii) just the second line (print statement) appear in the generated solution.

**Add-var anchor lines.** We also study *add-var* anchor lines, which return the sum of all variables in the function signature (converted to strings). For a function with inputs `var1` and `var2`, the add-var anchor lines are:

---

[3]Following Chen et al. [2021], we measure performance on HumanEval with *functional accuracy*: the fraction of programs that pass all of the test cases provided at the url: https://github.com/openai/human-eval.

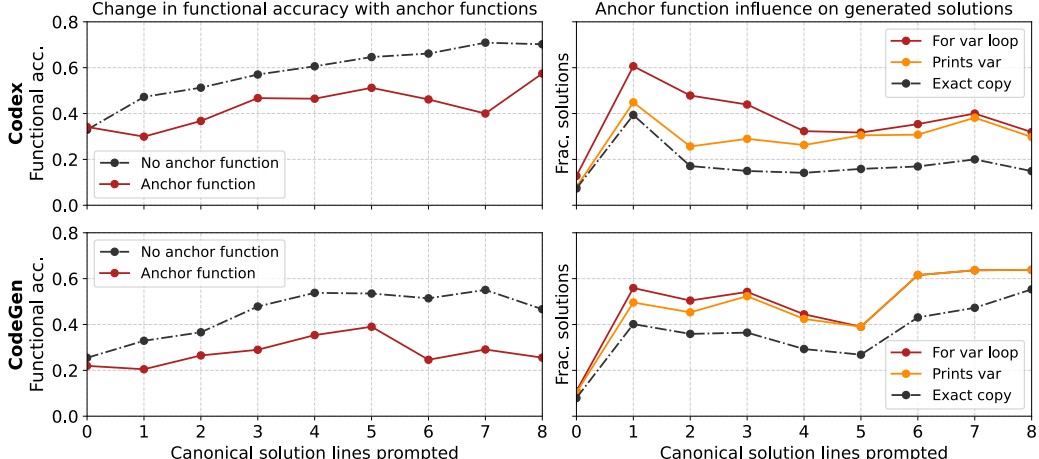

Figure 4: Results of the print-var anchoring experiment. **Left.** We measure the functional accuracy of Codex (top) and CodeGen (bottom) with no anchor function prepended (baseline acc) and with a print-var anchor function prepended (anchor acc), and find that prepending the anchor function consistently lowers accuracy. **Right.** We measure the influence of the anchor function on the generated solution by plotting the fraction of generated solutions that contain "`for var in `" from the print-var anchor prompt (for var loop), the fraction of generated solutions that include "`print(var) `" (prints var), and the fraction of generated solutions that output the anchor function verbatim without additional content (exact copy), as a function of the number of canonical solution lines added to the prompt.

```
tmp = str(var1) + str(var2)
return tmp
```

To study the influence of the add-var anchor lines on the solution, we measure how often `return tmp` appears in the generated solution.

**Print-var results.** In Figure 4, we show that prepending print-var anchor functions consistently lowers Codex and CodeGens' functional accuracies across different number of prompted canonical solution lines. We vary the number of canonical solution lines to study prompts of different difficulties; as the number of solution lines increases, the number remaining lines models must produce decreases.[4]

We additionally find that elements of anchor function often appear in both models' outputs, suggesting that code generation models adjust their solutions towards related solutions. In Figure 4, we see that Codex generates `for var` in 32%–61% of solutions when at least one line of the canonical solution is included, and generates `print(var)` in 26%–44% of solutions. CodeGen's behavior is qualitatively similar. Both models sometimes even incorporate the anchor lines into correct solutions; on Codex, the `for var` loop is used in a correct solution for 3%–11% of all outputs, while `print(var)` is used in a correct solution for 1%–9% of outputs.

**Control experiments.** One concern might be that models just outputs the anchor function verbatim, as in Section 3.3.1, but we find that this does not explain the full results—both models include anchor lines in many solutions that do not copy the anchor function verbatim. We also find that changing the name of the anchor function leads to only negligible changes; see Appendix A.1 for details.

**Add-var results.** We next consider results for add-var anchor lines. Full results for the add-var anchor prompts are presented in Appendix A.1 and are qualitatively similar to the print-var results.

One again, we find that prepending the anchor function consistently lowers functional accuracy. Moreover, the outputted solutions often include an anchor line. For example, Codex and CodeGen generate `return tmp` in 26%–46% and 13%–79% of solutions respectively, depending on how many canonical solution lines we prompt with. These results are not caused by models outputting the anchoring function verbatim: this only occurs between 7% and 12% of the time for Codex, and 4% and 12% for CodeGen. Overall, our findings suggest that code generation models can err by adjusting its output towards related solutions, when the solutions are included in the prompt.

---

[4]We filter out programs whose entire canonical solution would appear in the prompt; see Appendix A.1.

```
"""                                           """
Write a function that squares       Write a function that sums its
the sum of its inputs               inputs called product_plus_2
"""                                           """
def square_sum(x, y):               def product_plus_2(x, y):
    return x ** 2 + y ** 2              return x * y + 2
```

Figure 5: **Left.** Availability heuristic example where Codex mixes up the order of operations. The correct function signature (blue), `square_sum` matches the prompt. However, the incorrect function call (red) instead squares its inputs before summing them. The prompt is above the horizontal line, while the generated code is below. **Right.** Attribute substitution example where Codex relies on the function name to generate output. Codex correctly generates the desired function name (blue), but errs by using the function name instead of the prompt to generate the return statement (red).

### 3.3.3   Inspiration: Availability heuristic

We next draw inspiration from the *availability heuristic*: the tendency of humans to evaluate how frequently an example occurs based on how easy it is to recall. For example, Tversky and Kahneman [1973] find that humans tend to incorrectly report that there are more first words that start with "r" and "k" than have third letter "r" and "k", because the former quickly come to mind.

Using the availability heuristic as motivation, we hypothesize that code generation models may err by outputting solutions to related prompts that appear more frequently in the training set. To elicit this failure, we start with prompts that apply a unary operation before a binary operation (unary-first), then flip the order (binary-first). Programmers tend to apply unary operations first (e.g. when computing Euclidean distances or variances), so we conjecture that they appear more frequently on GitHub. We first check that flipping the order of operations decreases accuracy. Next, to test if code generation models instead outputs related prompts that occur more frequently in the training set, we measure whether code generation models instead output the unary-first solution.

We consider all 12 combinations of the binary operations sum, difference, and product, with unary operations square, cube, quadruple, and square root. Focusing on Codex,[5] we find that accuracy drops from 50% to 17% when flipping the order from unary-first to binary-first. Among combinations where flipping the order leads to error, we find that 75% of the binary-first outputs are the unary-first solution. We exhibit one such error in Figure 5: when prompted to square the sum of its inputs, Codex generates the correct function name (`square_sum` ), but reverses the order of operations. Our results suggest that Codex can err by outputting solutions to related, frequent prompts in the training set.

**Control experiments.** One worry is that the dip in performance is due the instructional nature of our prompts. We rule this out by evaluating Codex on prompts where the docstring appears beneath the function signature and is a definition rather than command, to more closely mimic some functions on GitHub. We obtain qualitatively similar results on these prompts, see Appendix A.4 for details.

### 3.3.4   Inspiration: Attribute substitution

Finally, we draw inspiration from *attribute substitution*: the human tendency to respond to a complicated question using a simpler, related question [Kahneman and Frederick, 2002]. For example, a professor when asked how likely a candidate is to be tenured, may instead respond with how impressive they found their job talk.

Using attribute substitution as inspiration, we hypothesize that Codex may use simple-but-incorrect heuristics to generate solutions. To elicit this failure, we add requests for conflicting function names to MathEquation prompts. For example, in Figure 5 we prompt Codex to write a program that sums its inputs called `product_plus_2` . We first check that adding conflicting function names decreases Codex's functional accuracy. Next, to test if Codex uses simple-but-incorrect heuristics to generate solutions, we check whether the generate solution matches the function name.

We evaluate Codex using 90 MathEquation prompts where the desired solution and requested function name differ. To construct prompts, we begin with a prompt that Codex originally solves

---

[5]We find that CodeGen often produces nonsensical solutions on the style of prompts used in Section 3.3.3, Section 3.3.4, and Section 5, so we focus primarily on Codex.

| Name location | Correct | Matches function name | Other error |
|---|---|---|---|
| No name | 100.0 | - | 0.0 |
| Docstring | 4.4 | 80.0 | 15.6 |
| Function signature | 4.4 | 70.0 | 25.6 |
| Name first | 4.6 | 51.7 | 43.7 |

Table 2: Results of the attribute substitution experiments. We report accuracy when we do not request a contradictory function name (no name), we request a function name in the docstring (docstring), in the function signature below the docstring (function signature), or above the docstring (name first). Overall, we find that Codex frequently generates solutions based on the function name.

(sum, difference, or product), then append a request for a specific, contradictory function name (see Appendix A.4 for full implementation details).

We report our experimental results in Table 2. When we request a conflicting function name, Codex's accuracy drops from 100% to only 4.4%-4.6%. This finding holds whether we request the function name in the docstring, write it in the function signature below the docstring, or write the function name over a simple description on the function. Moreover, for between 52% and 80% of prompts, Codex responds with the function specified in the function name. Our results indicate that Codex can err by using simple-but-incorrect heuristics to generate solutions.

## 4 GPT-3 Results

In this section, we extend our study from Codex to GPT-3. To test GPT-3 for failure modes, we try to faithfully reproduce and extend the anchoring experiment of Jacowitz and Kahneman [1995] and framing effect experiment of Tversky and Kahneman [1981].

**Anchoring.** As in Section 3.3.2 we study Section 3.3.2, we study anchoring: humans' tendency to insufficiently adjust their estimates away from an initial value [Tversky and Kahneman, 1974]. We largely replicate the anchoring study presented in Jacowitz and Kahneman [1995], but test the "davinci-001" version of OpenAI's GPT-3 instead of humans.

In their original experiment, Jacowitz and Kahneman asked students to estimate quantities such as the length of the Mississippi river in miles. They then asked new students to estimate the same quantities, but first gave them a upper or lower bound on the true answer (e.g. the Mississippi river is longer than 700 miles), which they call *anchors*. They find that students tend to underestimate the true quantity when prompted with the lower anchor, and overestimate it when prompted with the upper anchor.

We adapt the anchoring study from Jacowitz and Kahneman [1995] by finding the true answer for 14 of their 15 original questions[6], then computing upper and lower anchors by increasing and decreasing the true answer by a fixed percentage $p$. See Appendix B.1 for a full list of questions and true answers. As an example, if the actual answer is 2000 and $p$ is 50%, the upper anchor is 3000 and the lower anchor is 1000. We use this bound as an anchor, so that a typical prompt might be:

$$\underbrace{\text{The length of the Mississippi river (in miles) is greater than 1000.}}_{\text{anchor}}$$

What is the length of the Mississippi River (in miles)? Answer:

To study anchoring in GPT-3, we measure how prepending the anchor changes GPT-3's estimate. We categorize four potential changes: the estimate does not change, the estimate shifts towards the anchor, the estimate shifts away from the anchor, and the estimate is gibberish. We report the results in Table 3 for $p \in \{20\%, 50\%\}$. We find that GPT-3 routinely updates its estimate when an anchor is prepended, and tends to shift the estimate towards the anchor. We also find that while GPT-3's updated estimate sometimes matches the anchor exactly (67% of the time), it also often lands between the anchor and the original prediction, mirroring the behavior of humans.

Our replication has a few limitations. Like the original study our sample size is small, we construct prompts with templates, and many of the outputs—on average 41%—are gibberish. Nevertheless, our results suggest that GPT-3 incorporates the anchor during estimation.

---

[6]We omit a question asking for the number of bars in Berkeley, CA, since the answer is ambiguous

|       | **Anchor change to model output** | | | |
|-------|-----------|----------------|------------------|-----------|
| $p$   | No change | Towards anchor | Away from anchor | Gibberish |
| 20%   | 10.7      | 28.6           | 10.7             | 50.0      |
| 50%   | 14.3      | 42.9           | 10.7             | 32.1      |

Table 3: Results of the adaptation of the anchoring study from Jacowitz and Kahneman [1995] on GPT-3. We consider anchors that are 20% and 50% increases and decreases from the ground truth answer, and measure how often GPT-3's revised prediction does not change, shifts towards / away from the anchor, or is gibberish, aggregated across lower and upper anchors.

```
"""
Delete all files in an inputted directory that import all
of statsmodels, plotly, seaborn, and scipy
"""
import os
import re
def delete_all(directory):
    for root, dir, file in os.walk(directory):
        for f in files:
            with open(os.path.join(root, f), 'r') as fp:
                if re.search("import statsmodels", fp.read()):
                    os.remove(os.path.join(root, f))
```

Figure 6: **Left.** Example where Codex incorrectly deletes files. We prompt Codex to delete files containing all of statsmodels, plotly, seaborn, and scipy. Codex correctly iterates through all files in the inputted directory (blue), but then incorrectly deletes all files containing statsmodels (red), as attribute substitution suggests. **Right.** Plot describing the errors Codex makes as a function of the number of packages. We find that Codex often incorrectly deletes files if they contain any of the listed packages, and relies more on just the first package as the number of packages increases.

**Framing effect.** As in Section 3.3.1, we study the framing effect: predictable shifts in human responses when the same problem is framed in different ways. We largely replicate the framing experiment presented in Tversky and Kahneman [1981]: we compare GPT-3's responses to two equivalent decisions: choosing to either deterministically save (or let die) some fraction of a population, or to probabilistically save (let die) the whole population.

We measure the rate at which GPT-3 chooses the probabilistic option across different population sizes and different fractions / probabilities. See Section B.2 for full results. When using the probability in the original study, GPT-3 qualitatively mirrors humans: it chooses the probabilistic option far more frequently under the "not save" framing than under the "save framing". However, for higher probabilities, GPT-3 consistently chooses the probabilistic option for both framings; we conjecture that humans could exhibit similar behavior in this regime, since the probabilistic option is more certain. Overall, our results suggest that GPT-3 selects different options based on the framing, and could be a test-bed to identify qualitative human behaviors without running full human studies.

## 5  High-Impact Errors

We have shown how our framework helps us elicit failures of large language models. In this section, we use our framework to construct cases where Codex makes *high-impact errors*: harmful errors that are hard to undo. Specifically, we construct prompts where Codex incorrectly deletes files.

As in Section 3.3.4 we draw inspiration from attribute substitution: the tendency of humans to respond to a complex question with a simpler, related question. Using attribute substitution as motivation, we hypothesize that Codex may simplify complex expressions such as conjunctions. Instead of checking all components of a conjunction at once, it might "give up" and consider subsets of the components individually (e.g. checking for $A$ or $A \vee B$ instead of $A \wedge B$). To elicit this failure, we prompt Codex to delete files containing specific sets of package imports; see Figure 6 for an example. We measure how often Codex generates a simpler output that erroneously deletes files, as well as how often it produces the correct output. See Appendix C for additional details.

We test for two types of simpler outputs: code deleting all files containing first package in the set (i.e. $A$ instead of $A \wedge B$), and code deleting all files containing any package in the set (i.e. $A \vee B$

instead of $A \wedge B$). The latter operation is computationally simpler than checking if a file contains all packages, since Codex can delete a file whenever a single package in the set appears.

In Figure 6, we illustrate the breakdown of the errors Codex makes as a function of the number of package imports in the prompt. We find that Codex erroneously deletes files on at least 80% of prompts when the number of package imports is at least three, despite producing a correct output on 90% of prompts when the number of packages is at most two. Moreover, we find that Codex increasingly errs by using only the first package as the problem gets more challenging (i.e. the number of packages increases), as attribute substitution predicts.

**Control experiments.** To very that our findings generalize to different classes of realistic prompts, we test Codex on prompts containing a descriptive docstring beneath the function signature `delete_all_with_libraries(directory)`. We observe qualitatively similar results, though we find more instances of low-impact errors; see Appendix C for details. Overall, our results demonstrate how our framework can preemptively elicit high-impact errors, like erroneous deletions.

## 6  Discussion

In this work, we identify and test for classes of errors that open-ended generation systems can make, using cognitive biases as motivation. To do so, we generate hypotheses for potential qualitative failure modes, then construct transformations over prompts that elicit these failures. Our experiments uncover deficiencies of Codex, CodeGen, and GPT-3, and elicit high-impact errors that are challenging to undo. While we focus on a few specific failure modes, future work could apply our framework to uncover additional failures. Moreover, our framework queries systems as a black-box, so it could be used to quickly probe for errors in future systems as they are released.

Some of our results highlight how optimizing likelihood could be at odds with human intent. For example, over GitHub, programs may more often match their function signature than docstring (Section 3.3.3), or tend to complete to `pass` if the preceding function does (Section 3.3.1). Nevertheless, our results elicit qualitative errors regardless of the "correct" behavior (i.e. even when what is incorrect and correct flips), and demonstrate the importance of documenting qualitative failures.

The reliability challenges posed by the open-ended generation systems that we study sometimes also apply to classifiers. Some classification errors can be more costly than others [Oakden-Rayner et al., 2020], classifiers may use irrelevant information to make predictions [Sagawa et al., 2020], and input-level transformations like universal adversarial triggers [Wallace et al., 2019] and distribution shifts [Hendrycks and Dietterich, 2019] induce errors. However, while classification errors may be succinctly summarized with a confusion matrix, generation errors cannot, since each output appears infrequently. To tame the large output space, our transformations must induce categories of errors that we can reliably measure. Despite this additional constraint, we are able to construct *model-agnostic* transformations: we do not use the training data, model parameters, or even output logits. Our success in this restricted setting demonstrates the comparative brittleness of completion systems.

We present a method to systematically elicit errors from large language models. While we believe our work is important to understand model behavior, bad actors could exploit the errors we reveal (e.g. by deleting files on systems with a Codex back-end). Nevertheless, we introduce new robustness challenges for developers and identify misuses of these models, which we feel supersedes this risk.

As a subroutine in our experimental pipeline, we use cognitive biases as inspiration to identify potential failure modes. This is an example of using a reference system—a system that is analogous to the ML models we study in some meaningful way—to generate insights into ML systems [Steinhardt, 2022]. We use humans as the reference, focusing specifically on their susceptibility to cognitive biases. Other references, such as complex systems or evolution, may uncover new errors and insights. Moreover, ML systems could additionally err in ways that known systems do not, so it will also be useful to have *intrinsic* methods for characterizing model errors. Overall, our work underscores the need for more extensive testing of generative ML systems before their widespread deployment.

## Acknowledgements

We thank the anonymous reviewers, Ruiqi Zhong, Jean-Stanislas Denain, Aditi Raghunathan, Jessy Lin, and Lawrence Chan for feedback. This work was supported by NSF Award Grant no. 1804794.

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
