```
def common(l1, l2):
    ret = set()
    for el in l1:
        for var in [l1, l2]:
            print(var)
```
Anchor Function

```
def common(l1: list, l2: list):
    """Return sorted unique common elements for two lists.
    >>> common([1, 4, 3, 653, 5], [5, 1, 5, 9, 653, 121])
    [1, 5, 653]
    >>> common([5, 3, 2, 8], [3, 2])
    [2, 3]
    """
    ret = set()                          ret = set()
    for el in l1:                        for el in l1:
        for var in [l1, l2]:                 for e2 in l2:
            if el1 in var:                       if e1 == e2:
                ret.add(e1)                          ret.add(e1)
    return sorted(ret)                   return sorted(ret)
```

Figure 7: Actual example of how an anchor function impacts the generated solution. We construct the anchor function by taking the function signature from the HuamnEval prompt (blue), removing the docstring and variable typing, appending $n$ lines of the canonical solution (green), then adding anchoring lines (red). We prompt Codex with the anchor function, the HumanEval prompt, and the first $n$ lines of the canonical solution (above black line). The full canonical solution is on the right (green text, grey box). We see that the solution Codex generates (below black line) combines elements of the canonical solution (e.g. checks condition and adds to ret.), with the anchor function (e.g. for var loop).

# A    Additional Details and Results for Code Generation Experiments

In this section, we provide additional experimental details and results for the experiments in Section 3. We include additional details for anchoring (Appendix A.1), the availability heuristic (Appendix A.3), and attribute substitution (Appendix A.4).

## A.1    Anchoring

In this section, we include additional experimental details and results from the anchoring experiments in Section 3.3.2.

### A.1.1    Additional experimental details

**Filtering prompts for longer canonical solutions.** In Section 3.3.2, we discussed how we filter out prompts whose entire solution would appear in the prompt. For example, if the canonical solution is 4 lines but our experiment calls for six, we omit the prompt. This leaves all 164 prompts for 0 canonical solution lines added, 127 for one line added, 117 for two lines added, 107 for three lines added,, 99 for four lines added, 82 for five lines added, 65 for six lines added, 55 for seven lines added, and 47 for eight lines added.

**Additional prompt example.** In Figure 3, we showed an example prompt and output from our anchoring experiment. We expand on these results in Figure 7 by including the entire prompt, and the canonical solution as reference.

**Changing the anchor function name** We additionally study anchoring experiments where the name of the anchor function and function to be completed differ. This is different from the experiment in Section 3.3.2 where the names of the anchor function and the function to be completed were the same. Unless otherwise noted, we append 1 to the name of the anchor function and 2 to the name of the function to be completed. We propagate this change to other instances of the function name in the

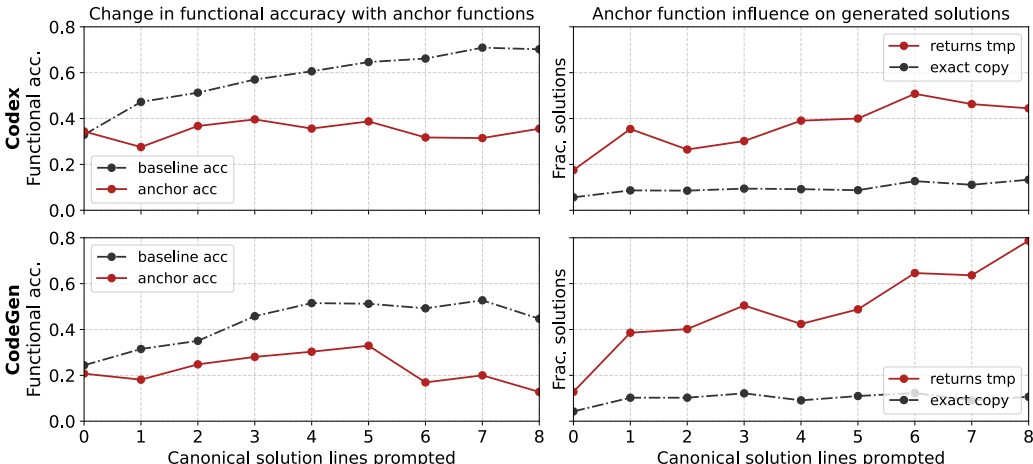

Figure 8: Results of the add-vars anchoring experiment. **Left.** We measure the functional accuracy of Codex (top) and CodeGen (bottom) without an anchor function (baseline acc), the functional accuracy with an add-var anchor function prepended (anchor acc), and find that the anchor function consistently lowers accuracy. **Right.** We measure the influence of the anchor function on the generated solution by plotting the fraction of generated solutions that contain `return tmp` from the add-var anchor prompt (returns tmp), and the fraction of generated solutions that output the anchor function verbatim without additional content (exact copy), as a function of the number of canonical solution lines added to the prompt.

function signature of and the docstring. However, all components of the prompts from Section 3.3.2 remain unchanged.

## A.2 Additional experimental results

In this section we provide additional experimental results, including the add-var anchor function results, tables containing numbers used to generate plots, and the results of our experiments where the anchor function and function to be completed have different names.

**Add-var results** We first exhibit the results of the add-var anchor line experiments described in Section 3.3.2. In Figure 8, we plot the functional accuracy of prompts with (baseline) and without (anchor) prepended anchor functions for both Codex and CodeGen, and find that while the baseline functional accuracy increases, the anchor functional accuracy remains roughly constant. Moreover, we see that both models adjust their output to related-but-incorrect solutions; in the same plot, we see that our test for the anchor, the presence of `return tmp` consistently appears in the generated solutions, while both anchor lines rarely appear together.

**Tabular results.** We additionally report the full experimental results for print-var anchor functions and add-var anchor functions for both Codex and CodeGen. Table 4, and the full experimental results for add-var anchor function sin Table 5. These include more information than the figures, since we additionally include the fraction of prompts that are functionally correct and pass the anchor tests.

**Control experiment: changing the function name.** In Figure 9 we plot the results of the print-var anchoring experiment where we append 1 to the function name in the anchor function, and 2 to the function name of the function to be completed (see Table 6 for numerical results). We plot the analogous add-var results in Figure 10 and include full numerical results in Table 7. Both results are nearly identical to the results where the function name is shared presented in Section 3.3.2, and suggest that the shared function name is not responsible for our anchoring results.

## A.3 Availability Heuristic

In this section, we augment Section 3.3.3 with additional additional availability heuristic experiments that use non-instructional prompts. These prompts give the correct function name, add in variables $x$

| Model | Sol. lines | Anc. acc. | Prints | P. + pass | For var | F.v. + pass | Copy | No anc. |
|-------|-----------|-----------|--------|-----------|---------|-------------|------|---------|
| | 0 | 34.1 | 7.9 | 0.0 | 12.8 | 1.8 | 7.3 | 32.9 |
| | 1 | 29.9 | 44.9 | 0.8 | 60.6 | 5.5 | 39.4 | 47.2 |
| | 2 | 36.8 | 25.6 | 1.7 | 47.9 | 7.7 | 17.1 | 51.3 |
| | 3 | 46.7 | 29.0 | 5.6 | 43.9 | 11.2 | 15.0 | 57.0 |
| CODEX | 4 | 46.5 | 26.3 | 5.1 | 32.3 | 6.1 | 14.1 | 60.6 |
| | 5 | 51.2 | 30.5 | 3.7 | 31.7 | 3.7 | 15.9 | 64.6 |
| | 6 | 46.2 | 30.8 | 1.5 | 35.4 | 3.1 | 16.9 | 66.2 |
| | 7 | 40.0 | 38.2 | 3.6 | 40.0 | 3.6 | 20.0 | 70.9 |
| | 8 | 57.4 | 29.8 | 8.5 | 31.9 | 10.6 | 14.9 | 70.2 |
| | 0 | 22.0 | 10.4 | 0.0 | 11.0 | 0.0 | 7.9 | 25.5 |
| | 1 | 20.5 | 49.6 | 0.0 | 55.9 | 1.6 | 40.2 | 32.9 |
| | 2 | 26.5 | 45.3 | 0.0 | 50.4 | 2.6 | 35.9 | 36.6 |
| | 3 | 29.0 | 52.3 | 0.0 | 54.2 | 0.9 | 36.4 | 47.8 |
| CODEGEN | 4 | 35.4 | 42.4 | 1.0 | 44.4 | 2.0 | 29.3 | 53.8 |
| | 5 | 39.0 | 39.0 | 1.2 | 39.0 | 1.2 | 26.8 | 53.5 |
| | 6 | 24.6 | 61.5 | 0.0 | 61.5 | 0.0 | 43.1 | 51.4 |
| | 7 | 29.1 | 63.6 | 3.6 | 63.6 | 3.6 | 47.3 | 55.1 |
| | 8 | 25.5 | 63.8 | 2.1 | 63.8 | 2.1 | 55.3 | 46.7 |

Table 4: Full results for the print-var anchor experiments, used to generate the plot in Figure 4. For different numbers of canonical solution lines (sol. lines), we report the functional accuracy when the anchor function is prepended (anc. acc.), the fraction of generated solutions that include `print(var)` (prints), the fraction of generated solutions that include `print(var)` and are functionally correct (p. + pass), the fraction of generated solutions that include `for var in` (for var), the fraction of generated solutions that include `for var in` and are functionally correct (f. v. + pass), the fraction of solutions that are exactly the anchor function (copy), and the functional accuracy without the anchor function prepended (no anc.).

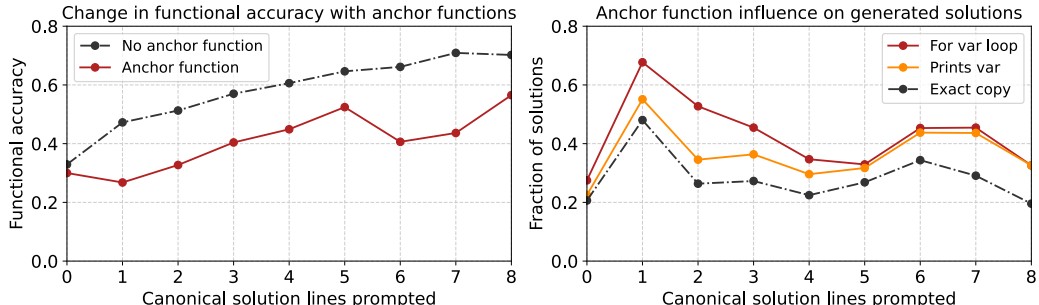

Figure 9: Results of the print-var anchoring experiment on Codex, where we append 1 to the name of the anchor function and 2 to the name of the function to be completed.

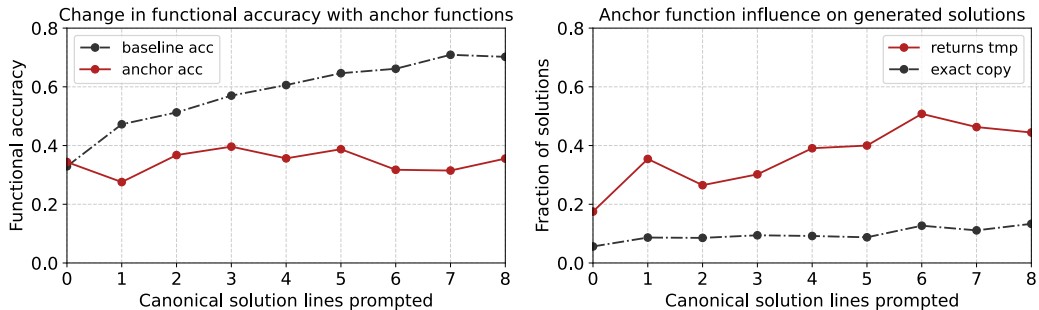

Figure 10: Results of the add-var anchoring experiment on Codex, where we append 1 to the name of the anchor function and 2 to the name of the function to be completed.

| Model | Sol. lines | Anc. acc. | Rets. temp | Rets. tmp + passes | Verbatim | No anc. acc. |
|---|---|---|---|---|---|---|
| | 0 | 32.3 | 7.9 | 0.6 | 4.3 | 32.9 |
| | 1 | 33.1 | 29.1 | 1.6 | 7.1 | 47.2 |
| | 2 | 39.3 | 29.9 | 2.6 | 7.7 | 51.3 |
| | 3 | 46.7 | 26.2 | 2.8 | 6.5 | 57.0 |
| CODEX | 4 | 38.4 | 35.4 | 1.0 | 7.1 | 60.6 |
| | 5 | 41.5 | 37.8 | 0.0 | 8.5 | 64.6 |
| | 6 | 36.9 | 44.6 | 3.1 | 12.3 | 66.2 |
| | 7 | 40.0 | 43.6 | 0.0 | 7.3 | 70.9 |
| | 8 | 40.4 | 42.6 | 0.0 | 8.5 | 70.2 |
| | 0 | 20.7 | 12.8 | 0.6 | 4.3 | 24.4 |
| | 1 | 18.1 | 38.6 | 0.0 | 10.2 | 31.5 |
| | 2 | 24.8 | 40.2 | 0.0 | 10.3 | 35.0 |
| | 3 | 28.0 | 50.5 | 0.9 | 12.1 | 45.8 |
| CODEGEN | 4 | 30.3 | 42.4 | 0.0 | 9.1 | 51.5 |
| | 5 | 32.9 | 48.8 | 0.0 | 11.0 | 51.2 |
| | 6 | 16.9 | 64.6 | 1.5 | 12.3 | 49.2 |
| | 7 | 20.0 | 63.6 | 0.0 | 9.1 | 52.7 |
| | 8 | 12.8 | 78.7 | 2.1 | 10.6 | 44.7 |

Table 5: Full results for the add-var anchor experiments, used to generate the plot in Figure 8. For different numbers of canonical solution lines (sol. lines), we report the functional accuracy when the anchor function is prepended (anc. acc.), the fraction of generated solutions that include `return tmp` (rets. tmp), the fraction of generated solutions that include `return tmp` and are functionally correct (rets. tmp + passes), the fraction of solutions that are exactly the anchor function (Verbatim), and the functional accuracy without the anchor function prepended (no anc. acc.).

| Sol. lines | Anc. acc. | Prints | P. + pass | For var | F.v. + pass | Copy | No anc. |
|---|---|---|---|---|---|---|---|
| 0 | 30.0 | 22.5 | 0.0 | 27.5 | 1.9 | 20.6 | 32.9 |
| 1 | 26.8 | 55.1 | 0.8 | 67.7 | 4.7 | 48.0 | 47.2 |
| 2 | 32.7 | 34.5 | 2.7 | 52.7 | 5.5 | 26.4 | 51.3 |
| 3 | 40.4 | 36.4 | 5.1 | 45.5 | 7.1 | 27.3 | 57.0 |
| 4 | 44.9 | 29.6 | 4.1 | 34.7 | 5.1 | 22.4 | 60.6 |
| 5 | 52.4 | 31.7 | 2.4 | 32.9 | 2.4 | 26.8 | 64.6 |
| 6 | 40.6 | 43.8 | 1.6 | 45.3 | 3.1 | 34.4 | 66.2 |
| 7 | 43.6 | 43.6 | 7.3 | 45.5 | 7.3 | 29.1 | 70.9 |
| 8 | 56.5 | 32.6 | 6.5 | 32.6 | 6.5 | 19.6 | 70.2 |

Table 6: Full results of the print-var anchoring experiment on Codex where we append 1 to the name of the anchor function and 2 to the name of the function to be completed. These numbers are used to generate the plot in Figure 9.

| Sol. lines | Anc. acc. | Rets. temp | Rets. tmp + passes | Verbatim | No anc. acc. |
|---|---|---|---|---|---|
| 0 | 34.4 | 17.5 | 1.2 | 5.6 | 32.9 |
| 1 | 27.6 | 35.4 | 1.6 | 8.7 | 47.2 |
| 2 | 36.8 | 26.5 | 2.6 | 8.5 | 51.3 |
| 3 | 39.6 | 30.2 | 1.9 | 9.4 | 57.0 |
| 4 | 35.6 | 39.1 | 0.0 | 9.2 | 60.6 |
| 5 | 38.8 | 40.0 | 1.2 | 8.8 | 64.6 |
| 6 | 31.7 | 50.8 | 1.6 | 12.7 | 66.2 |
| 7 | 31.5 | 46.3 | 0.0 | 11.1 | 70.9 |
| 8 | 35.6 | 44.4 | 0.0 | 13.3 | 70.2 |

Table 7: Full results of the add-var anchoring experiment where we append 1 to the name of the anchor function and 2 to the name of the function to be completed. These numbers are used to generate the plot in Figure 9

and $y$, and add the description below the function signature, but keep all other experimental details from Section 3.3.3 constant. An example prompt is as follows:

```
def square_sum(x, y):
    #function squares the sum of its inputs
```

Codex achieves higher accuracy with this prompt than the prompt from Section 3.3.3; it achieves an accuracy of 54.1%. However, the unary-first bias remains: 27.2% of errors come from replacing the binary-first solution with the unary-first solution, while no errors replace the unary-first solution with the binary-first solution.

### A.4 Attribute substitution

In this section, we provide more details on how we generate prompts for the attribute substitution experiment in Section 3.3.4.

**Prompts in Section 3.3.4.** We consider two types of MathEquation prompts for our experiments in Section 3.3.4. First, we consider prompts that include the function name in the docstring:

```
"""
Write a function that computes the [operation] of its inputs called [name]
"""
```

And second, we consider prompts that already include the function name.

```
"""
Write a function that computes the [operation] of its inputs
"""
def [name]
```

We consider names of the form [operation]_plus_[number]. We test sum, difference, and product for operations, and consider the integers between 0 and 5 and powers of ten between 10 and 10000 for the possible numbers for 90 total prompts in each setting. These are the prompts we use to report numbers in Table 2.

**Control experiment: non-instructional prompt.** We next test non-instructional prompts, where the prompt includes the correct function name, variables $x$ and $y$, and a description below the function signature. Other experimental details from Section 3.3.4 remain constant. An example prompt is as follows:

```
def product_plus_2(x, y):
    #returns the sum of its inputs
```

## B  Additional Details and Results for GPT3 Experiments

In this section, we include additional details and results on experiments described in Section 4. We focus on the anchoring results in Appendix B.1, and the framing effect results in Appendix B.2

### B.1  Anchoring

In this section, we provide more details for the replication study of Jacowitz and Kahneman [1995] described in Section 4. Specifically, we outline the prompts that we use in the study along with GPT-3's outputs.

In Table 8, we show the prompts we use from the Jacowitz and Kahneman [1995] study along with the true answer, then the lower and upper anchors using $p$ of 50%. We additional study $p = 20\%$ to generate the results in Table 3. We find the true answers for meat consumption[7], distance from San Francisco to New York[8], the height of the tallest redwood[9], the number of female professors at

---

[7] https://thehumaneleague.org/article/meat-consumption-in-the-us (as of 2017)
[8] https://www.distance24.org/New%20York%20City/San%20Francisco
[9] https://www.livescience.com/28729-tallest-tree-in-world.html

| Question | Actual | Low. anc (50%) | Up. anc (50%) |
|---|---|---|---|
| Length of the Mississippi River (in miles) | 2350 | 1175 | 3525 |
| Height of Mount Everest (in feet) | 29032 | 14516 | 43548 |
| Amount of meat eaten per year by the average American (in pounds) | 144 | 72 | 216 |
| Distance from San Francisco to New York City (in miles) | 2569 | 1284 | 3854 |
| Height of the tallest redwood (in feet) | 380 | 190 | 570 |
| Number of United Nation members | 193 | 96 | 290 |
| Number of female professors at the University of California, Berkeley | 256 | 128 | 384 |
| Population of Chicago (in millions) | 2.7 | 1 | 4 |
| Year the telephone was invented | 1876 | 938 | 2814 |
| Average number of babies born per day in the United States | 10267 | 5134 | 15400 |
| Maximum speed of a house cat (in miles per hour) | 30 | 15 | 45 |
| Amount of gas used per month by average American (in gallons) | 656 | 328 | 984 |
| Number of state colleges and universities in California | 23 | 12 | 34 |
| Number of Lincoln's presidency | 16 | 8 | 24 |

Table 8: Prompts we use from the Jacowitz and Kahneman [1995], with the researched true answer, along with the lower and upper anchors with anchor adjustment 50%.

Berkeley[10], the population of Chicago[11], the year the telephone was invented[12], the number of babies born per day in the United States[13], the maximum speed of a house cat[14], and the amount of gas used per month by the average American[15] at the URLs listed in the footnotes. We do not prompt GPT-3 to estimate the Number of bars in Berkeley, CA unlike the original study, since we could not find a reliable answer.

## B.2 Framing effect

In this section, we test GPT-3 with an expansion of framing effect study from Tversky and Kahneman [1981]. In their original experiment, Tversky and Kahneman asked people to choose between two treatment options: certainly saving some fraction of the population (e.g. certainly saving 200 / 600), or probabilistically saving all of the population (saving all 600 with probability 1/3). They then compare peoples' responses to the identical choice framed in terms of death: choosing between some fraction of the population certainly dying (400 / 600 die) or probabilistically letting the whole population die (2/3 chance everybody dies). For both framings, the number of people that live and die in expectation remains constant.

We run this experiment on the "davinci-002" version of GPT-3 using Tversky and Kahneman [1981]'s prompt formatting. Specifically, for the *save framing* we prompt GPT-3 with the following four-lined prompt:

```
Imagine 600 people are affected by a deadly disease. Choose Option A or Option B
Option A: Exactly 200 people will be saved.
Option B: 1/3 probability that 600 people will be saved, and 2/3 probability that
    no people will be saved.
Answer: Option
```

---

[10] https://www.dailycal.org/2020/04/02/female-faculty-faces-challenges-despite-increase-in-uc-berkeley-gender-diversity/

[11] https://en.wikipedia.org/wiki/Demographics_of_Chicago

[12] https://www.sciencemuseum.org.uk/objects-and-stories/ahoy-alexander-graham-bell-and-first-telephone-call

[13] https://www.babycenter.com/pregnancy/your-body/surprising-facts-about-birth-in-the-united-states_1372273 (as of 2019)

[14] https://www.petfinder.com/cats/cat-behavior-and-training/how-fast-cats-run-how-high-cats-jump

[15] https://www.fool.com/investing/2017/01/14/heres-how-much-gasoline-the-average-american-consu.aspx (As of 2017)

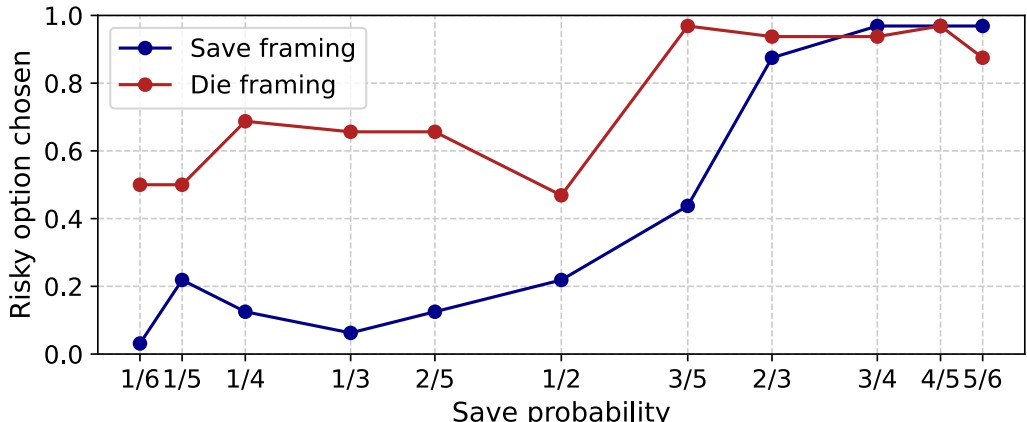

Figure 11: Fraction of the time the risky option is chosen as a function of the save probability for both save framing and die framing. Around the regime of the original experiment (save probability = 1/3), GPT-3 rarely chooses the risky option with the save framing, but does so more often with the die framing. However, for higher save probabilities, both tend to choose the risky framing. This might match humans; intuitively, for higher save probabilities, the risky framing is less risky.

| | Save probability range | | | | |
|---|---|---|---|---|---|
| Prompt framing | Less than 0.5 | 0.5 | Greater than 0.5 | All | Tversky and Kahneman [1981] |
| Save framing | 11.3 | 21.9 | 84.4 | 45.4 | 28 |
| Die framing | 60.0 | 46.9 | 93.8 | 74.1 | 78 |

Table 9: Results of the framing effect experiment on GPT-3. For the save and die framinigs, we report the average probability (all), the average over different ranges of probabilities, and the original numbers reported in Tversky and Kahneman [1981].

For the *death framing*, we use an analogous four-lined prompt.

```
Imagine 600 people are affected by a deadly disease. Choose Option A or Option B
Option A: Exactly 400 people will die.
Option B: 1/3 probability that nobody will die, and 2/3 probability that 600 people
    will die.
Answer: Option
```

In these prompt, the *population size*, or total number of affected people is 600, and the *save fraction*, or fraction of people certainly saved is 1/3. The original study only considers these numbers, but we additionally test population sizes of 60, 300, 900, 1200, 1500, 3000, and 6000 and all save fractions that have denominator less than seven and are reduce (i.e. 1/2, not 2/4). We test the rate at which GPT-3 selects the *risky option* or option which probabilistically lets everyone die, for both the save framing and die framing. To avoid the confounding influence of the position of the risky option and the label, for each prompt framing, population size, and save fraction we set the risky option to both option A and B, and put it both first and second. This gives us a total of 704 prompts.

In Figure 11 plot the fraction of the time the risky option is chosen as a function of the save probability for both the save and die framing, and results aggregated over save probability ranges in Table 9. Our results show that averaged over all probabilities, our results are qualitatively similar to the results from Tversky and Kahneman [1981]: in the original paper with the save framing people choose the risky option 28% of the time compared to 45% for GPT-3, and with the death framing choose the risky option 78% of the time compared to 74% for GPT-3. Around the regime of the original experiment (save probability = 1/3), GPT-3 rarely chooses the risky option with the save framing (11.3%), but does so more often with the die framing (60.0%). However, for higher save probabilities, both tend to choose the risky framing. This might match humans; intuitively, for higher save probabilities, the risky framing is less risky; e.g. the human has a 80% chance of saving everyone. This experiment highlights both how language models might mirror cognitive biases of humans, and how they could

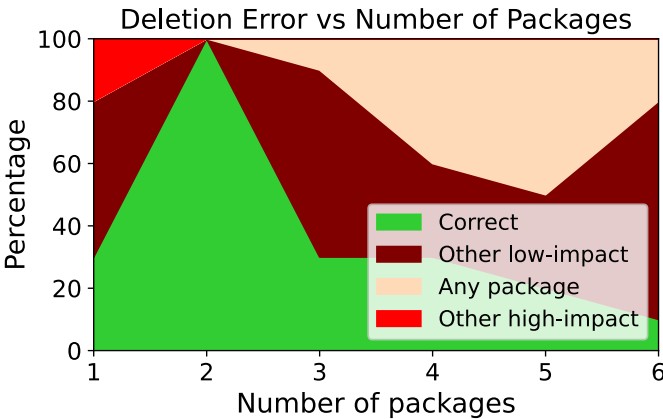

Figure 12: Plot describing the errors Codex makes as a function of the number of packages for the docstring deletion experiment.

be useful for studying how humans make decisions—our results suggest that a human study where the save probability is higher might have qualitatively different conclusions.

## C  Additional deletion experimental details

In this section, we include additional experimental details for the deletion experiments

**Additional prompt details.**  Recall that in Section 5 we exhibit how Codex can make *high-impact errors* by inadvertently deleting files. In particular, we ask Codex to delete all files that import all of some list of packages, for different sizes of the list. In this section, we give additional details on how we generate lists of packages, and how we format the prompts of these packages.

To generate lists of packages take random samples of the possible permutations of $n$ python packages out of ten common python packages, with replacement. We sample 10 permutations of sizes 1-6 to generate six total prompts. We select from the 10 packages listed at https://www.kdnuggets.com/2021/03/top-10-python-libraries-2021.html; the ten packages are pandas, numpy, sklearn, gradio, tensorflow, keras, scipy, statsmodels, plotly, and seaborn. One prompt to codex is given in Figure 6. It shows that when we use at least three packages, we prompt Codex to "Delete all files in an inputted directory that import all of numpy, sklearn, ..., and gradio". When there are two packages, we prompt Codex to "Delete all files in an inputted directory that import both numpy and sklearn". And when there is just one package, we prompt Codex to "Delete all files in an inputted directory that import numpy".

We use manual evaluation to categorize the results. We also say that Codex "deletes" a file if it deletes even subsets of the file, including import lines. This occurred in a small minority of generated outputs.

As a control experiment, we first test if descriptive prompts rather than instructional prompts show similar results to those in Section 5. To do so, we largely replicate the deletion experiments from Section 5; i.e. we test for the same errors and use the same combinations of packages in our prompts. However, we specify that the function name is `delete_all_with_libraries`, and the docstring appears below the prompt. Our prompts look like:

```
def delete_all_with_libraries(directory):
    #deletes all files in an inputted directory that import...
```

**Additional experimental results**  In this section, we report the results of our descriptive-prompt control experiment. These results are in in Figure 12. Overall, we still see many high-impact errors (the "any package" region), but see far more low-impact errors, like endless import strings, compared to Section 5. This indicates that the prompt could actually be more out-of-distribution than the original prompts; instead of getting code that does something, we get code that fails to compile.