# OpenReview forum: "Capturing Failures of Large Language Models via Human Cognitive Biases"
_NeurIPS.cc/2022/Conference — NeurIPS 2022 Accept_

### Official Review · Reviewer_RQFZ · 2022-07-11

**Rating:** 7
**Confidence:** 4
**Soundness:** 3 good
**Presentation:** 4 excellent
**Contribution:** 2 fair

**Summary:**

Authors highlight the fragile nature of the GPT-3 style code generation models. Inspired from cognitive biases research, authors design different manual input transformations (prompt tuning) where they add some irrelevant information to the model. Using experiments with Codex model, they show that model's code generation performance drop significantly when they fed modified input to the model. It's worth highlighting that the proposed transformations are generally non-adversarial in nature.

**Questions:**

- One can argue that framing effect is by design and is not an unexpected behavior. Based on the GPT-3 paper's in-context learning experiments, we expect model to imitate the behavior of the functions provided in the context. Hence, when a preceding function throws `NotImplementedError`, it's probably reasonable for model to generate `NotImplementedError` as *one of the answers* (assuming we sample multiple answers with a temperature).
- Which decoding algorithm are you using for generation? Is it greedy decoding or are you using some sampling algorithm? Further, for code generation tasks, a common practice is to use sampling and report pass@K metric as it reduces variance in the results. Have you observed the same accuracy drop with pass@K ?
- Do you have any counterexamples i.e. transformations where you did not see any accuracy drop?
- "Open-ended systems" is a very broad term. Since this paper mainly focuses on the code generation task, I would recommend to use "open-ended generation" instead of using “open-ended systems”.

**Typos**:
- L41: that that
- L111: that that
- L140: prictable -> predictable
- L298: able construct -> able to construct


References:
1. Li, Yujia et al. “Competition-Level Code Generation with AlphaCode.” ArXiv abs/2203.07814 (2022)

**Limitations:**

I don't see any major concerns related to negative societal impact.

**Strengths And Weaknesses:**

Strengths:

- Proposed transformations are very intuitive and are well-motivated by prior research in cognitive science.

Weaknesses:

- While the transformations described in the paper are intuitive, some of them are hard to scale. In particular, one has to design such transformations manually for a given programming language setup. Further, authors have not provided an easy to use list of transformations (or dataset version) which other researchers can reuse for future experiments.
- Prior work [1] show that problem description, context are crucial to execution accuracy.  This paper's experiments also underline that execution accuracy highly depends upon the prompt (context) fed to the model and minor changes in the context can lead to big drop in execution accuracy. I am not sure that this paper offers new insights to the community.
- Authors crafted these prompts manually and show that all of these transformations lead to significant accuracy drop. It's not clear 1) how many different prompts they tried for a given experiment. 2) Have they reported results for all transformations or only for those transformations which lead to accuracy drop.
- Paper is missing crucial details related to how the code was generated. In particular, it's not clear which decoding method (greedy, sampling with temperature, etc.) is used to generate the data. Without such crucial details, it's hard to replicate the results presented in the paper.

---

> ### Author Response · Authors · 2022-08-02
> **Response to Reviewer RQFZ**
>
> Thanks for your review; we appreciate that you found our transformations intuitive.
>
> **On crafting prompts manually**
>
> One concern you highlighted is that we crafted prompts manually, giving us too much power to solicit behaviors. For the framing effect and anchoring experiments, we generate our prompts by concatenating or applying transformations to HumanEval prompts (a pre-existing dataset) in an effort to deliberately limit our influence on the results. While we do write prompts for the deletion, availability heuristic, and attribute substitution experiments with fewer constraints, we did not manually engineer prompts to obtain results. And for the rebuttal, we ran experiments with prompts suggested by 3Gtn (so no author input), and got qualitatively similar results. We hope that this gives you more confidence in the robustness of our findings.
>
> **On our contribution**
>
> A second concern is that our contribution is simply demonstrating that *“execution accuracy highly depends upon the prompt (context) fed to the model and minor changes in the context can lead to big drop in execution accuracy”*, which you claimed does not *“offer new insights to the community”*. However, our main contribution is in measuring **how** the output changes when we apply our targeted transformations [109 - 116] (e.g. adjusting outputs towards related function in Section 3.3.2). This is especially important in the structured output setting that we study since simply showing that models perform worse with unwanted context is not informative, as models can fail in many qualitatively different ways [21-25].
>
> **On resources for future research**
>
> Finally, a third concern is that there is not a dataset of transformations to use for future experiments, and that we don’t extend the transformations to other languages. We provided code in the supplementary material to generate prompts, but  to ease reproducibility we have created an anonymous repository with all of the prompts used available here: [https://anonymous.4open.science/r/capturing-failures-of-llms-prompts-172C/README.md](https://anonymous.4open.science/r/capturing-failures-of-llms-prompts-172C/README.md). Moreover, most of our transformations could easily be adapted to other languages (e.g. converting docstrings / function signatures but preserving the logic), allowing for broad applicability to code models.
>
> **Responses to individual questions**
>
> *It's not clear 1) how many different prompts they tried for a given experiment. 2) Have they reported results for all transformations or only for those transformations which lead to accuracy drop. Do you have any counterexamples i.e. transformations where you did not see any accuracy drop?*
>
> We report all of the transformations that we tried, and observe varying degrees of accuracy drop; for example, in the anchoring experiment the drop ranges from 0 points (0 canonical solution lines prompted) to roughly 30 points (7 lines prompted). We hope that the additional prompts we try in the response to 3Gtn also help address this concern.
> ***
> *Which decoding algorithm are you using for generation? Is it greedy decoding or are you using some sampling algorithm?*
>
> We use greedy decoding; while we alluded to this in the checklist, we explicitly articulate this in the revision.
> ***
> *"Open-ended systems" is a very broad term. Since this paper mainly focuses on the code generation task, I would recommend to use "open-ended generation" instead of using “open-ended systems”.*
>
> Good idea; we replaced *open-ended systems* with *open-ended generation systems* in the revision.
> ***
> *Typos*
>
> We fixed all of the typos you noted in the revision; thanks for finding them!

---

> > ### Comment · Reviewer_RQFZ · 2022-08-10
> > **Thanks for additional experiments**
> >
> > Thanks for answering my questions and conducting CodeGen experiments. With new experiments, the findings presented in the paper are  convincing. I have adjusted my scores accordingly.  I would encourage to 1) moving Codegen experiments to the main paper (I would personally prefer Codegen over OpenAI API simply because it's open-source and is reproducible) 2) open-source the modified prompts for reproducibility.

---

### Official Review · Reviewer_3Gtn · 2022-07-11

**Rating:** 7
**Confidence:** 4
**Soundness:** 3 good
**Presentation:** 4 excellent
**Contribution:** 3 good

**Summary:**

This paper proposes to use human cognitive biases as inspiration for types of errors that large language models may make, as a means of finding and catching such errors. They run experiments on four such cognitive bias-inspired errors on code generation with Codex models. They also include two additional sets of smaller experiments, one with GPT-3 with a pre-existing human cognitive bias experiment, and another with Codex on code deletion as a means of emphasizing high-impact errors.

**Questions:**

- Most of my specific questions about the experiments are covered in the above section.
- What decoding configuration was used for Codex/GPT-3? If temperature was used (i.e. non-greedy decoding), it should be reported and potentially run with multiple runs. The checklist mentions "deterministic rollout", but explicit clarification of the decoding configuration would be helpful.

**Limitations:**

The authors have framed the work in terms of being helpful to catch errors of large language models, which I think is reasonable. The work does have experiments testing the propensity for models to indiscriminately delete files, but I believe the experiments are conducted within a reasonable and contained setting.

**Strengths And Weaknesses:**

As a foreground, my view on the use of human cognitive biases as a motivation in this paper is that it is a useful and valid approach for coming up with potential error categories, but ultimately the results in the paper can be interpreted and have value independent of the associated human cognitive biases. Because of this, and my relative unfamiliarity with human cognitive bias work, I will focus my comments on evaluating the experiments primarily from the point of a view of a machine learning researcher.

Strengths:

- The paper is exceedingly clear, well-written and easy to follow. The examples of the prompt formats is extremely welcome and papers that do prompt-based experiments would do well to follow this example.
- The experiments are well-structured and well-motivated. While I do have concerns regarding some of the experimental setup (described below), the logical flow from human cognitive biases to a proposed equivalent model experiment to the metrics being measured are generally clear, reasonable, and cover the questions a critical reader might have (e.g. also measuring copying of the irrelevant propmt).

Weaknesses:

- Experimental setup / evaluation. My major concern with this paper is the experimental design, and whether the experiments correspond to the research question being asked. I will discuss my concerns with each of the experimental setups here. This section includes both what I consider weaknesses as well as questions/additional results I would like to see.

1) Framing.

I have an issue with the choice of "irrelevant preceding function" and the interpretation of model behavior, in that I do not share the authors' interpretation that the prepended functions are irrelevant. For instance, given a prompt:

```
def add(x, y):
    """Adds numbers"""
    raise NotImplementedError()

def multiply(x, y):
    """Multiplies numbers"""
```

I *would* expect the model to output `raise NotImplementedError()`. This looks like a file that is work-in-progress, and to my mind the model is correctly extrapolating from the prompt. In the authors' words, I do not believe this is "semantically irrelevant information". A better example (and this overlaps somewhat with anchoring) would be something like:

```
def divide(x, y):
    """Divides numbers"""
    assert y != 0
    return x / y

def multiply(x, y):
    """Multiplies numbers"""
```

In this case, if the model outputs `assert y!= 0`, then we can conclude that the model is actually using the prepended function in an erroneous/unhelpful way. Put another way, if the experimenter wants the model to generate functional code, they should to prompt *with* functional code. This should be a pretext for all experimental involving functional accuracy.

2) Anchoring.

I think the setup and results in this experiment are clearer than for framing. One concern I have is with the confounding factor of the anchor function having the same name as the function to be completed. An experiment with a similar name instead (e.g. common1 vs common2) would clarify this result.

3) Availability.

The authors conjecture without evidence that "Programmers tend to apply unary operations first", and use this as the basis for their conclusion that Codex has learned to output "related prompts that occur more frequently in the training set". From the results presented, we can take away that the model appears more biased toward unary-first solutions, but it seems difficult to draw conclusions relating to the training set without further evidence.

I also have a milder concern regarding the formatting of "docstrings" in this and the attribute substitution experiments. Docstrings should go under the function signature and describe the function. The "docstrings" in these two experiments are provided above the function signature, and written in the form of an instruction "Write a function that...", which is not how docstrings are used in practice, and therefore should not be part of the expected behavior of the model.

4) Attribute substitution.

My primary concern with this experimental setup is that it introduces contradictory function names and docstrings (see also the discussion above), and asserts that the model ought to take the dosctrings as the ground truth and that the function names are misleading. It is not clear to me that this has to be that case. I could conversely argue that in practice, documentation updates often lag behind code changes, and that the function name is a more reliable source-of-truth than docstrings. Under this interpretation, what the results in Table 2 show is that the model mostly "correctly" adheres to the function name and is only mildly affected by the "incorrect" docstring information. Even in this lens, I think the results are still valuable, because the conflicting information appears to introduce a set of "Other errors". I recommend the authors revisit and rethink the design and interpretation of this experiment.

(Admittedly, this is confounded in the "docstring" sub-experiment, since the only source of information is the dosctring.)

6) Deletion

This suffers from the same issue as the attribution substitution experiments, though not to the same degree (`delete_all` can be reasonable seen to be context dependent). That said, I wonder if the results would be different if the function were named more helpfully (`delete_all_with_libraries`).

In summary, while I think this approach for finding model errors is valid and worthwhile, the current experiments fall short of demonstrating the biases in a convincing manner. Addressing these issues would make a meaningful difference to my review.

- Scale of the experiments. The experiments tend to be on a small scale with a small-ish number of examples. Particularly given the potential of models to be sensitive to the exact phrasing of prompts, this is a moderate but not large concern.
- Lack of evaluation on open models. As the authors acknowledge, the Codex model is private and many details regarding the architecture and training are not public. This makes it difficult to ascertain, for instance, that the models have no had additionally curated data to avoid some of these errors (i.e. a more naively model may perform even worse than reported in the paper). While I acknowledge that this paper is primarily focused on Codex and duplicating experiments on an additional model with a separate setup is significant additional work for the authors, I think it is fair to say that experiments solely on a set of non-public models with many unknowns does hurt the scientific contribution of the paper and calls into question whether these results generalize. Even a small set of experiments on a similar more public model (e.g. CodeGen) that demonstrate equivalency of results would help assuage the reader.

---

> ### Author Response · Authors · 2022-08-02
> **Response to Reviewer 3Gtn**
>
> Thanks for your detailed review! We appreciate how you find our approach for finding model errors *“valid and worthwhile”*, our experiments *“well-motivated and structured”*, and our presentation clear. Based on your helpful suggestions, we present experiments with the modifications to prompts you suggest, experiments on a second model you recommended (CodeGen), and then address your more specific questions below. We hope that with these results, you’ll consider raising your score.
>
> **Experiments with suggested prompts**
>
> Your primary concern was about whether our experimental setup answered the research questions we asked, either because the prompt formatting leads to strange results, or the results are the “expected” behavior of Codex. In response, you proposed experiments with different prompts that you would find more convincing. We implemented many of these, and have included them in the revision of the paper. In particular, we ran the following:
> 1. Docstring reformatting: we reran the availability heuristic, attribute substitution, and deletion experiments with the docstring underneath the function signature to address the concern that putting the docstring above *“is not how docstrings are used in practice, and therefore should not be part of the expected behavior of the model.”* We added all of the results to Appendix A.6.1, but to summarize:
>     * The availability heuristic results are qualitatively similar; there are no errors flipping unary-first to binary-first, but are errors shifting binary-first to unary-first.
>     * The deletion experiments (using the suggested `delete_all_with_libraries` function name) still often incorrectly delete files (up to 50\% of the time), but more frequently output erroneous code that does not delete files.
>     * The attribute substitution results still default to the function name instead of the docstring 52\% of the time, and only defer to the docstring 5\% of the time. However, we find that 22\% of the time Codex erroneously combines the operation in the docstring with the constant in the prompt, suggesting that regardless of whether Codex “should” use the docstring or the function name, it still errs by relying on the other.
> 2. Anchoring experiment: to address *“the confounding factor of the anchor function having the same name as the function to be completed”*, we reran the anchoring experiments by naming anchor function `[fn]1` and function to be completed `[fn]2`, following your suggestion that *“An experiment with a similar name instead (e.g. common1 vs common2 would clarify the result”*. We put the results in Appendix A.6.1, and find that they’re nearly identical to the existing print-var and add-vars results.
>
> **Experiments on open-sourced models**
>
> You also highlighted how we don’t evaluate on open-sourced models, and suggested that *“[e]ven a small set of experiments on a similar more public model (e.g. CodeGen) that demonstrate equivalency of results would help assuage the reader.”* Following your suggestion, we reran the framing effect and anchoring experiments on the 6B version CodeGen, and report the results in Appendix A.7. At a high level, we find that this model is indeed more susceptible to our transformations than Codex. Specifically, we observe the following:
> 1. Framing effect (Table 11): for each framing line, we observe a larger percentage drop in functional accuracy on CodeGen compared to Codex. Moreover, the rate at which the framing line appears often increases: for example, the rate at which `print(“Hello world!”)` is outputted increases from 32.6\% on Codex to 54.5\% on CodeGen.
> 2. Print-vars and Add-vars anchoring experiments (Figures 13 &14, Tables 12 & 13): We observe the same qualitative results on CodeGen and Codex; accuracy decreases when anchor functions are prepended, and elements of the anchor function tend to appear in the generated solutions. Both effects are larger than on Codex.
>
> We will integrate these results with the Codex results in the main body using the extra page in the final version.
>
> **Response to additional questions**
> ***
> *What decoding configuration was used for Codex/GPT-3?*
>
> We use greedy decoding for both; we updated the paper to feature this more prominently.
> ***
> *From the results presented, we can take away that the model appears more biased toward unary-first solutions, but it seems difficult to draw conclusions relating to the training set without further evidence.*
>
> To get some empirical gauge of how frequent these are among math equations in the training data, we count the number of functions in numpy and numpy.linalg that compose unary and binary operations. Of these, we find that 77% of such functions apply the unary operation first. While this isn’t the same as the uniform distribution over GitHub, we still think it is an informative sample.
> ***
> *For the framing effect and availability heuristic experiments, the model is following the expected behavior.*
>
> See response to all reviewers.

---

> > ### Comment · Reviewer_3Gtn · 2022-08-10
> > **Response**
> >
> > I have adjusted my score in response to the authors' response and additional experiments. While I am still concerned about the small number of experiments/data, their additional experiments have made their findings more convincing.
> >
> > My two additional recommendations to the authors are:
> >
> > 1) Better integrate the additional results into the main paper. Adding to the appendix for the response is fine, but I would recommend the readers reorganize the presentation of results as some of the experiment variants are worth highlighting
> > 2) Make the prompts/experimental data, and ideally the experimental scripts, publicly available for easy reproducibility and inspection.

---

> ### Author Response · Authors · 2022-08-09
> **Thoughts on our Rebuttal?**
>
> Thanks again for your review; based on it we ran the experiments you suggested on different models and prompt formats, which we hope resolves the potential weaknesses you identify. Do these new experiments help address your major concerns?

---

### Official Review · Reviewer_Nt74 · 2022-07-11

**Rating:** 6
**Confidence:** 4
**Soundness:** 3 good
**Presentation:** 3 good
**Contribution:** 3 good

**Summary:**

This paper explores the robustness of large language models through the lens of 4 human cognitive biases - framing, anchoring, attribute substitution, and availability. Results show that OpenAI's Codex program synthesis model is not robust to changes in prompts that are inspired by these 4 biases.

**Questions:**

**Major**
* Figure 4 (right) - explain why “fraction of solutions” containing anchor function lines is so low when # of canonical solution lines prompted is 0
* 3.2.4 - relationship to attribute substitution cognitive bias is a stretch. Not very convincing that confusing the model with conflicting function name vs. desired behavior is an example of model erring “by using simple-but-incorrect heuristics to generate solutions”. Recommend replacing this with examples from Section 5 and removing section 5, which doesn't seem to warrant a separate section
* Would be nice to see more cognitive bias methods applied to GPT, since it shows diff/similarity between LM and human

**Minor**
* Footnote 2 - please share how many examples were filtered out
* Figure 4 - clarify whether this is on humaneval only
* Print-var results + add-var results sections - recommend rounding results to whole numbers. Tells same story, easier to digest
* Table 3 - would be interesting to plot anchor values vs. prediction (e.g. expect to see as upper bound anchor increases, GPT-3 prediction increases) to clearly demonstrate anchoring effect

**Limitations:**

Yes

**Strengths And Weaknesses:**

**Strengths**
* Well-written and easy to understand for the most part (especially introduction and some figures)
* Categorizes a few types of errors for program synthesis models and develops interesting evaluations for them (i.e. functional accuracy + custom evaluations for each error type)
* Experiments demonstrate that Codex, and likely other language model-based program synthesis models, are not robust to certain types of perturbations
* Example of anchoring applied to GPT is especially interesting, since it shows a similarity between how language models (LMs) and humans behave

**Weaknesses**
* Paper seems to over-generalize contribution - e.g. claims to study "failure modes of large language models", but contributions are rather specific to prompt robustness. I would expect a paper with this general of a title and abstract to study more errors more comprehensively (e.g. anaphora resolution, entailment, long context, etc.). I recommend narrowing down the claims made in this paper
* Contribution is relatively simple (changing prompts and measuring robustness). Could expand scope by increasing types of cognitive biases tested, or more experiments on GPT
* Missing some key details for reproducibility and understanding
  * Needs details on how functional accuracy is checked (e.g. environment, open source code)
  * Figure 3 - unclear what is going on. Maybe add full example and/or more explanation in caption
  * Figure 5 - clarify what colors mean, and what is input vs. generated
  * Section 5 - what examples are used here, and how many? How were examples generated and evaluated?
* Section 3.2.3 on Availability bias unsubstantiated
  * L213-214 - “Our results suggest that Codex can err by outputting solutions to related, frequent prompts in the training set.”. Need to show evidence that unary operations applied before binary in training set to make this claim. Recommend sampling training set and counting unary-first vs. binary-first

---

> ### Author Response · Authors · 2022-08-02
> **Response to Reviewer Nt74**
>
> Thank you for your helpful comments and suggestions; we appreciate that you found our evaluations interesting and our work clear and easy to understand. Below we provide the additional experiments that you requested and then address your more specific questions.
>
> You mentioned that we could *expand scope by increasing the types of cognitive biases tested, or more experiments on GPT*. To do so, we test for the framing effect by expanding the classic experiment from Tversky and Kahneman (1981). We observe qualitatively similar decisions from GPT-3 to the humans in the original study, include this in Appendix A.5 of the revised version, and will promote this to the main body using the additional page in the final version. We include a more detailed description of this experiment in the *GPT-3 Framing Experiment* comment after this response and hope that if this result and the additional experiments other reviewers suggest strengthen the scope, you’ll consider increasing your score.
>
> We address your specific questions below:
> ***
> *Paper seems to over-generalize contribution - e.g. claims to study "failure modes of large language models", but contributions are rather specific to prompt robustness.*
>
> While many of the experiments relate to prompt robustness, not all of them do. For example, our deletion experiments show how Codex can erroneously delete files with standard prompting, and in the availability heuristic experiments we show that Codex can mix up standard compositions of operations. We think studying additional failure modes is an important direction for future work.
> ***
> *Needs details on how functional accuracy is checked (e.g. environment, open source code)*
>
> We evaluate functional accuracy on HumanEval using the code from the original Codex paper: [https://github.com/openai/human-eval](https://github.com/openai/human-eval) . We made this clearer in the revision.
> ***
> *Section 3.2.3 on Availability bias unsubstantiated. Need to show evidence that unary operations applied before binary in training set to make this claim. Recommend sampling training set and counting unary-first vs. binary-first*
>
> We find that randomly sampled pages from GitHub nearly always don’t have any composed unary-first or binary-first operations. To get some empirical gauge of how frequent these are among math equations, we count the number of functions in numpy and numpy.linalg that compose unary and binary operations. Of these, we find that 77\% of such functions apply the unary operation first. While this isn’t the same as the uniform distribution over GitHub, we still think it is an informative sample.
> ***
> *Figure 4 (right) - explain why “fraction of solutions” containing anchor function lines is so low when # of canonical solution lines prompted is 0*
>
> Great question; we don’t know for sure, but see this phenomena on CodeGen (requested by Reviewer 3Gtn) as well. One hypothesis is that without any code from a valid solution to the prompt (i.e. just the anchor lines), there isn’t enough relevant information in the anchor function for Codex to use it in the target completion.
> ***
> *Figure 3 - unclear what is going on. Maybe add full example and/or more explanation in caption*
>
> We included a full example in Appendix A.1 and hope this helps! We now refer to the appendix in the caption for clarity.
> ***
> *Figure 5 - clarify what colors mean, and what is input vs. generated”: clarified*
> Everything above the line is the prompt, while everything below the line is generated. Correct elements of the generation are in blue, while erroneous elements are in red. We’ve clarified this in the revision.
>
> ***
> *Section 5 - what examples are used here, and how many? How were examples generated and evaluated?*
>
> We forgot to include the link to the corresponding section in the appendix; sorry about that! We’ve added the link in the revised version.
> ***
> *Figure 4 - clarify whether this is on humaneval only*
>
> It is only on HumanEval.
> ***
> *Print-var results + add-var results sections - recommend rounding results to whole numbers. Tells same story, easier to digest*
> Updated!
>
> ***
> *Footnote 2 - please share how many examples were filtered out*
>
> We’ve added this detail in the appendix; the number of examples filtered varies based on the number of canonical solution lines added.

---

> > ### Author Response · Authors · 2022-08-02
> > **GPT-3 Framing Experiment**
> >
> > We now describe how we test for the framing effect in GPT-3 by expanding the classic experiment from Tversky and Kahneman (1981); full details are in Appendix A.5. The original experiment measures how the framing of options impacts how people choose between probabilistic and deterministic options that are equivalent in expectation. Specifically, options are framed in terms of saving people versus letting people die (see lines [122 - 127]). We take the original prompts from Tversky and Kahneman (1981), but vary the fraction of the time everyone is saved in the probabilistic option and overall population size, letting us aggregate over 704 prompts.
> >
> > We find that GPT-3, like humans, adapts its decision based on the framing: it chooses the probabilistic option more often when framed in terms of dying (74.1% vs 78% for humans) compared to saving lives (45.4% vs 28% for humans). Moreover, inspired by your suggestion for our GPT-3 anchoring experiment, we study how the fraction of the time everyone is saved changes the results [Figure 9]. We find that there is a phase change in results: for the save framing and low save probabilities (in the regime of the original experiment) GPT-3 nearly always chooses the deterministic option, but for higher save probabilities (no existing human experiment) nearly always chooses the probabilistic option. This is intuitive; the probabilistic option with high probability saves everyone in this case, and it would be interesting to see if humans behave similarly.
> >
> > Here are some example prompts:
> >
> > Save framing:
> > ```
> > Imagine 600 people are affected by a deadly disease. Choose Option A or Option B
> > Option A: Exactly 200 people will be saved.
> > Option B: 1/3 probability that 600 people will be saved, and 2/3 probability that no people will be saved.
> > Answer: Option
> > ```
> >
> > Die framing:
> > ```
> > Imagine 600 people are affected by a deadly disease. Choose Option A or Option B
> > Option A: Exactly 400 people will die.
> > Option B: 1/3 probability that nobody will die, and 2/3 probability that 600 people will die.
> > Answer: Option
> > ```

---

> ### Author Response · Authors · 2022-08-09
> **Thoughts on Our Rebuttal?**
>
> Thanks again for your review; based on it we ran the GPT-3 experiments you suggested, which we hope resolves the potential weaknesses you identify. Do these experiments and our responses to your clarification points help assuage your concerns?

---

### Official Review · Reviewer_Ey5D · 2022-07-12

**Rating:** 6
**Confidence:** 3
**Soundness:** 3 good
**Presentation:** 3 good
**Contribution:** 3 good

**Summary:**

The paper aims to evaluate generation models in text using qualitative categories - patterns that demonstrate systematic errors as opposed to individual instance errors. The main case-study is performed on OpanAI’s Codex, where the authors find that Codex does predictable mistakes which are dependent on (a) how the input prompt is framed (b) outputs are closer to anchors, or (c) biased towards instances that are similar to highly frequent training samples.

The paper also claims that the proposed framework, which uses the concept of human cognitive biases, to elicit hypotheses of errors, can unearth high-impact errors – which for Codex amounts to generations that can delete files.


**Questions:**

- The issues highlighted in Anchoring might be related to in-context prompting. [Recent work](https://arxiv.org/pdf/2202.12837.pdf) highlights how labels are the most crucial elements in the context. In the current scenario, it is possible that the model thinks of these anchor functions as the corresponding labels which it tries to incorporate into the solution.
While I agree this behavior is not ideal, we probably should consider it as a feature and not a bug. As anchoring might have beneficial usage in controlling the output in zero- or few-shot.
With this in mind, probably it’s better to test if the issues of anchoring persist if the model is clarified which part of the context is a demonstration (the anchor part here), and which part is the query?

- Some of the attacks seem to be highly out-of-distribution prompts - For example, calling a summing function as `product_plus_2` is probably an extreme example. As such, it would be more interesting if cognitive bias patterns can be used to find adversarial prompts whose differences are more imperceptible?


**Ethics Review Area:**

["I don’t know"]

**Limitations:**

There is no clear section dedicated to the limitations of the current work - maybe the authors should think about situations where cognitive bias framework would fail to elicit errors.
Also, are there cases where output of certain prompt attacks can never be anticipated?

**Strengths And Weaknesses:**

Strengths:

- Essentially the paper attempts to understand how large language models are sensitive to the context provided - do they focus on irrelevant information or distractors? And can their failures be categorized into frequent modes. The novelty claimed here is that the paper analyzes these failures from the lens of cognitive biases.

- The paper can be seen as an extension of Checklist (Ribeiro et al.). Here, the authors rely on cognitive bias literature to design adversarial prompts with expected outputs.

- Unearthing high-impact errors is an important use-case.

Weaknesses:

- The nature of the prompts or the output tendencies (mimicking patterns from training corpora), etc. have been studied in the literature before on different applications. For example, checking for sensitivity to context and the noise present in them is a known approach. The contribution that this paper makes is probably in collating these testing strategies and clubbing them under the cognitive bias framework.

- The connection to cognitive biases, while interesting, at times seem forced. Nevertheless, it is an interesting way to look at adversarial examples and might be beneficial as it summarizes some important patterns of failure. Particularly, patterns such as framing (noise in context), and anchoring are interesting as they have wide applicability.

---

> ### Author Response · Authors · 2022-08-02
> **Response to Reviewer Ey5D**
>
> Thanks for your feedback on your paper! We appreciate that you thought that the *“connection to cognitive biases summarizes some important points of failure”*, that our *“framing + anchoring results have wide applicability”*, and that our work *“unearthing high-impact errors is an important use-case”*.
>
> You point out that *“the nature of the prompts or the output tendencies (mimicking patterns from training corpora), etc. have been studied in the literature before on different applications” which makes the contribution “collating these testing strategies and clubbing them under the cognitive bias framework”*. However, our main contribution is in measuring **how** the output changes in the structured output setting when we apply our targeted transformations [109 - 116] (e.g. adjusting outputs towards related functions in Section 3.3.2) . This is especially important  in the structured output setting compared to classification (which covers most existing work) since simply showing that models perform worse with unwanted context is not informative, as models can fail in many qualitatively different ways [21-25].
>
> We address your specific questions below:
> ***
> *The issues highlighted in Anchoring might be related to in-context prompting. Recent work highlights how labels are the most crucial elements in the context.*
>
> Our anchoring results are actually in contrast to the results from Min et al. The Min et al. paper argues that the labels *aren’t* important for classification; they show that if you randomize the label for the prompted few-shot examples, performance does not decrease by much. In contrast, our transformations consistently lead to errors, and moreover induce specific qualitative failure modes (e.g. adjusting outputs towards related solutions, erroneously deleting files) that we can measure.
> ***
> *Some of the attacks seem to be highly out-of-distribution prompts - For example, calling a summing function as `product_plus_2` is probably an extreme example.*
>
> This is true; while some of the transformations are more natural (e.g. deleting files, applying unary + binary operations), others are more extreme. However, the extreme nature of some of our examples allows us to better isolate the effect of different parts of the prompt; in this case, it helps us isolate the impact of the function name.
> ***
> *Are there cases where output of certain prompt [transformations] can never be anticipated?*
>
> Yes: we suspect there are ways ML systems can make errors that do not resemble known systems. We discuss this a bit in lines 308 - 312, and identify developing system-intrinsic methods for characterizing model errors as an important direction for future work.

---

> ### Author Response · Authors · 2022-08-09
> **Thoughts on our Rebuttal?**
>
> Thanks again for your review; we responded to the points you raised, and also ran new experiments to assuage the concerns of some of the other reviewers. Does our rebuttal help address your feedback?

---

### Official Review · Reviewer_mPku · 2022-07-14

**Rating:** 5
**Confidence:** 4
**Soundness:** 2 fair
**Presentation:** 3 good
**Contribution:** 2 fair

**Summary:**

In this paper, the authors assess the quality and reliability of open ended systems such as GPT-3 and Codex from the perspective of cognitive biases. The authors primarily focus on the application of code generation and design examples that elicit commonly made errors by these systems for the task of code generation and completion. The authors primarily focus on the 4 different set of cognitive biases: (1) framing effect; (2) anchoring bias; (3) availability heuristic and (4) Attribute substitution.

The main contributions:
- Creation of framework that contains hypothesis and prompts that extend existing cognitive bias methodology. The hypothesis and examples created can measure the model sensitivity to these transformations. The framework is also capable of discovering high impact errors and other types of errors made by the model.


**Questions:**

- it would be better if the authors could add more explanation about how the drop in performance related to cognitive biases.

**Limitations:**

The authors address limitations of systems such as GPT-3 and Codex. However, there is no mention of limitations of designing prompts

**Strengths And Weaknesses:**

Strengths & Weakness:
- The paper studies an important problem of robustness of large scale LM's such as Codex and GPT-3 from the task of of code generation. Using cognitive bias as a way to measure robustness of these models is an interesting and novel approach.
-  Results in the paper show that by manipulating exhibits different learning effects and thus resulting in significant drops in performance. However, claiming these to be exhibits of cognitive biases seems to be a bit of a stretch. Most of the errors discovered could be novel from the perspective of code generation but prior literature in the area of robustness and fairness and prompt engineering shown that these large scale model are susceptible to unwanted context added to prompts

---

> ### Author Response · Authors · 2022-08-02
> **Response to Reviewer mPku**
>
> Thank you for your review! We appreciate that you find our approach *“interesting and novel”*, and that you find the problem we study interesting.
>
> One concern that you raised is that *claiming drops in performance to be exhibits of cognitive biases is a stretch”*. We do not intend to make this claim; instead we use cognitive biases to *“construct a transformation over prompts that largely preserves semantics, but that we expect will elicit a specific failure mode”*, while using the high-level methodology from work in heuristics and biases (Section 3.2 [109 - 116]).
>
> A second concern is that the empirical results don’t offer novelty beyond prior literature showing that *“these large scale models are susceptible to unwanted context added to prompts.”*
> However, our main contribution is in measuring **how** model output changes when we apply our targeted transformations [109 - 116] (e.g. adjusting outputs towards related functions in Section 3.3.2). This is especially important in the structured output setting that we study since simply showing that models perform worse with unwanted context is not informative, as models can fail in many qualitatively different ways [21-25].  Uncovering qualitative failure modes systematically is core to the novelty of this work.
>
> We address individual questions below:
> ***
> *It would be better if the authors could add more explanation about how the drop in performance related to cognitive biases.*
>
> For the experiments on Codex and new experiments CodeGen, cognitive biases are just used to motivate transformations; we don’t claim that the drop in performance relates to the cognitive bias. For experiments on GPT-3 we closely track original cognitive bias studies, and observe similar failures to humans.

---

> ### Author Response · Authors · 2022-08-09
> **Thoughts on our Rebuttal?**
>
> Thanks again for your review; we responded to the points you raised, and also ran new experiments to assuage the concerns of some of the other reviewers. Does our rebuttal help address your feedback?

---

### Author Response · Authors · 2022-08-02
**Response to all reviewers**

We thank all of the reviewers for their constructive feedback. All reviewers had positive feedback on our paper, saying that our approach is *“interesting and novel”* (mPku) and *“valid and worthwhile”* (3Gtn), our transformations are *“intuitive and well-motivated by prior research in cognitive science”* (RQFZ), our framing and anchoring results *“have wide applicability”* (Ey5D), and our evaluations are *“interesting”* (Nt74), and *“well-structured and well-motivated”* (3Gtn).

In response to explicit reviewer suggestions, we’ve added additional experiments with different prompts and additional models. Specifically, we added:
1. Additional Codex experiments: we run experiments with prompts requested by 3Gtn renaming functions in the anchoring experiments, and a change in docstring formatting for availability heuristic, attribute substitution, and deletion experiments. The results were all qualitatively similar, see Appendix A.6 for details.
2. CodeGen experiments: based on a suggestion by 3Gtn, we run our framing effect and anchoring experiments on the 6B parameter public CodeGen model, and find that on CodeGen, the qualitative failures we test for appear even more frequently than on Codex. See Appendix A.7 for details
3. Additional GPT-3 experiment: We add an additional experiment testing the framing-effect on GPT3 requested by Nt74 (see Appendix A.5). We expand on the framing effect experiment of Tversky and Kahneman [1], and find that GPT-3 exhibits qualitatively similar behavior to the humans in the original study.
We hope these suggested experiments highlight the robustness of our results, and help assuage reviewer concerns.

Multiple reviewers suggested that the framing effect and attribute substitution results were expected; in particular, when maximizing for likelihood models should e.g. output `raise NotImplementedError` when the preceding function does so, or should use the description in the function name over the docstring. However, we note that at a minimum our results highlight how the model behaves inconsistently; regardless of whether the “true” objective is to output `raise NotImplementedError` or the actual function, the model errs sometimes by doing the reverse. More importantly, our experiments document cases where likelihood and user-intent might diverge, which is especially important as these models are deployed in the wild.

We address specific reviewer feedback in individual sections.

[1] Amos Tversky and Daniel Kahneman. The framing of decisions and the psychology of choice. Science, 211(4481):453–458, 1981.

---

### Meta-Review · Area_Chair_HH4f · 2022-08-27

**Recommendation:** Accept
**Confidence:** Less certain

**Metareview:**

This paper aims to qualitatively categorize errors by large language models for generation tasks (i.e., summarization, program synthesis), drawing inspiration from 4 human cognitive biases: framing effect, anchoring bias, availability heuristic and attribute substitution. The work includes case studies on Open AI's Codex and GPT-3, and demonstrates that Codex makes predictable mistakes based on the framing of the input prompt, outputs that are closer to anchors and instances that are similar to frequent training examples. The paper uses the proposed framework to elicit high-impact errors. The paper is interesting and is clearly presented, the experiments are well-designed. The rebuttal includes additional experiments suggested by the reviewers with different prompts and additional models.

**Award:**

No

---

### Decision · Program_Chairs · 2022-09-14

Accept